



# Low frequency variability in North Sea and Baltic Sea identified through simulations with the 3-d coupled physical-biogeochemical model ECOSMO

Ute Daewel[1], Corinna Schrum[1,2]

[1]Helmholtz Centre Geesthacht, Institute of Coastal Research, Max-Planck-Str. 1, 21502 Geesthacht, Germany
[2]Geophysical Institute, University of Bergen and Hjort Centre for Marine Ecosystem Dynamics, Allegaten 41, 5007 Bergen, Norway

*Correspondence to*: Ute Daewel (ute.daewel@hzg.de)

**Abstract.** Here we present results from a long-term model simulation of the 3d coupled ecosystem model ECOSMO II for a North and Baltic Sea setup. The model allows both multi-decadal hindcast simulation of the marine system and specific process studies under controlled environmental conditions. Model results have been analysed with respect to long-term multi decadal variability in both physical and biological parameters with the help of empirical orthogonal function (EOF) analysis. The analysis of a 61-year (1948-2008) long hind cast reveals a quasi-decadal variation on salinity, temperature and current fields in the North Sea in addition to singular events of major changes during restricted time frames. These changes in hydrodynamic variables where found to be associated to changes in ecosystem productivity that are temporally aligned with the timing of reported "regime shifts" in the areas. Our results clearly indicate that for analysing ecosystem productivity spatially explicit methods are indispensable. Especially in the North Sea a correlation analysis between atmospheric forcing and primary production (PP) reveals significant correlations for NAO and wind forcing for the central part of the region, while AMO and air temperature are correlated to long-term changes in the southern North Sea frontal areas. Since correlations cannot serve to identify causal relationship we performed scenario model runs with perturbing the temporal variability in forcing condition emphasizing specifically the role of solar radiation, wind and eutrophication. The results revealed that, although all parameters are relevant for the magnitude of PP in the North Sea and Baltic Sea, the dominant impact on long-term variability and major shifts in ecosystem productivity was introduced by modulations of the wind fields.

## 1 Introduction

Long term variations and major changes in ecosystem dynamics occur throughout all trophic levels and have earlier been reported on in a number of studies for both the North Sea and Baltic Sea system (Beare et al., 2004; Beaugrand and Ibañez, 2000; Clark and Frid, 2001; Lynam et al., 2017; Möllmann et al., 2000; Schlüter et al., 2014; Selim et al., 2016; Thurow, 1997; Weijerman et al., 2005; Wiltshire and Manly, 2004). A majority of those studies have been thereby focussing on potential "regime shifts" RS ("Changes in marine system function that are relatively abrupt, persistent, occurring at a large spatial scale, observed at different trophic levels and related to climate forcing." deYoung et al., 2004). Such major changes throughout all trophic levels were e.g. reported for the North Sea and Baltic Sea System at the end of the 1980s (Alheit et al., 2005; Österblom et al., 2007; Weijerman et al., 2005). Beaugrand (2004) reviewed studies addressing RSs in the North Sea. He reported on studies considering temporal changes in single species abundance and vital rates throughout all trophic levels, system productivity and species composition within trophic levels or feeding guilds. By combining these studies with time series information on hydro-meteorological conditions for the same time periods Beaugrand (2004) hypothesised three different drivers for persistent changes in the North Sea ecosystem, i.) a change in the local hydro-meteorological forcing, ii.) a displacement of oceanic biogeographical boundaries, and iii.) an increase in oceanic inflow into the North Sea. Dippner et al. (2012) compared potential "regime shifts" in the North Sea and Baltic Sea and could associate the inter-annual variability and RSs in the Baltic Sea to changes in the atmospheric forcing only, while for the North Sea he found combined



influences from the atmospheric and the Atlantic forcing to be most likely responsible for inter-annual variations in ecosystem dynamics. In fact, many studies could relate variations in the ecosystem to variations in atmospheric variables and indices, such as NAO, SST and wind (Alheit et al., 2005; Beaugrand and Kirby, 2010; Edwards et al., 2010) but also to modification in the anthropogenic forcing such as fisheries or nutrient loads (Österblom et al., 2007). Nonetheless, the

identification of causal relationships and underlying processes is difficult based on in situ observations only, due to the complexity in identifying the relative relevance of single factors (Clark and Frid, 2001) and also the inhomogeneous characteristics of the datasets, which are often relatively short and lack the spatial diversity in regional ecosystem components.

However, understanding the relevance of environmental factors for ecosystem dynamics pioneers the identification of

environmental indicators for long-term variations and RSs. "Indicators are proxies for complex phenomena and can be used to reflect the provision of a service and how it is changing over time." (Hattam et al., 2015) Hence, the identification of potential indicators is of major relevance for both marine ecosystem understanding and management. Since bottom-up processes play a major role for long term variations in functioning of many regional marine ecosystem, and the North Sea and Baltic Sea system in particular (Daewel et al., 2014; Frank et al., 2007), understanding processes impacting net primary

productivity form the basis for indicator definition. To overcome limitation of observational based analysis, coupled physical-biological ecosystem models are valuable tools that provide spatially explicit long-term datasets of lower trophic level production (Daewel and Schrum, 2013). Additionally, these kinds of models allow further a clear analysis of environmental factors and underlying mechanisms, since the former are explicitly prescribed in the model formulation and setup. Additionally, specific scenarios can be applied by artificially modulating the forcing parameters to test hypothesis and

indicators.

Here we analysed further the 61-year long simulation (1948-2008), which was earlier presented by Daewel and Schrum (2013). The length of the simulation period allows identification of long-term changes in the environment and in primary production in the North Sea and Baltic Sea. Here, we aim at exploring key long-term variation in relevant environmental variables and the potential of different methods to derive environmental indicators describing the hydrodynamic and

biogeochemical environment. We evaluate the potential of aggregated hydrodynamic, atmospheric and large-scale climatic indicators to explain modelled primary production variability. Finally, we utilize the model to simulate specific scenarios to test the indicators for causal relations to interannual variations in simulated primary production.

## 2 Methods

### 2.1 ECOSMO II model description

ECOSMO II (ECOSystem Model, Daewel and Schrum, 2013; Schrum et al., 2006a) is a 3d fully coupled physical-biogeochemical model. The long-term simulation of lower trophic level ecosystem dynamics with ECOSMO II was presented and validated in Daewel and Schrum (2013). The hydrodynamic core of the coupled model system is a mature and in detail validated (e.g. Janssen et al., 2001; Schrum, 2001) 3D baroclinic coupled sea-ice model based on the version of the HAMSOM (HAMburg Shelf Ocean Model) presented first by Schrum (1997) and Schrum and Backhaus (1999). The model

is a free surface model and allows for variable bottom layer thickness; hence it resolves a realistic bathymetry. The model uses semi-implicit methods (Backhaus and Hainbucher, 1987), which allows for a relative large model time step of 20 min. In contrast to the earlier model version described by Schrum and Backhaus (1999), we use here a second order Total Variation Diminishing (TVD) scheme, namely the 2nd order Lax-Wendroff, which was made TVD by a superbee limiter (e.g. Harten, 1997) for the advection of all scalar properties. Its implementation and the consequences for ecosystem





dynamics are in more detail described by Barthel et al. (2012). The model equations are solved on a staggered Arakawa-C-grid for the North Sea and Baltic Sea, with a horizontal resolution of 6 nm (1 nm=1852 m) and 20 vertical levels, whereof the upper 40 m have a 5m resolution to resolve stratification. The model has earlier been used to investigate seasonal and interannual to decadal variations of stratification and have been found to successfully reproduce the latter in the North Sea

(Janssen et al., 2001; Schrum et al., 2000).

The biogeochemical processes in ECOSMO II were simulated using 16 state variables to resolve ecosystem dynamics by a functional group approach (Fig. 2). The model estimates two zooplankton functional groups, three phytoplankton groups, the nitrogen, phosphorus and silicon cycle, oxygen, detritus, biogenic opal, dissolve organic matter, and three sediment groups. The model equations, setup and a model validation for a 61 year model hind cast integration were presented in detail by

Daewel and Schrum (2013) who found the model able to reproduce temporal and spatial variability of primary and secondary production of the North Sea and Baltic Sea on intra- and interannual up to decadal time scales. The model was validated using nutrient data only, because of the better availability, reliability and comparability of nutrients in in-situ observations to model data compared to biomass estimates.

Atmospheric boundary conditions are required at the air-water interface and were taken from the NCEP/NCAR re-analysis

data (Kalnay et al., 1996). Sea surface elevation including the major tidal constituents as well as salinity and nutrients were prescribed at the open boundaries to the North Atlantic (see figure 1). For the remaining ecosystem variables and temperature a Sommerfeld radiation condition is applied at the open ocean boundaries (Orlanski, 1976). Additionally, river runoffs and nutrient loads are given at the land ocean boundary from a collection of different data sources. For more details on data sources and handling and a complete description on the simulation setup please consults Daewel and Schrum (2013).

**2.2 Statistical Methods**

The advantage of model-derived data is their spatially explicit characteristics, which also makes it difficult to analyse the data and to distinguish major modes of variability. A widely used method in climate and ocean science is the empirical orthogonal function analysis (EOFs), a statistical method to identify dominant modes in multidimensional data fields (e.g. Storch and Zwiers, 1999; Venegas, 2001). Here the method is used to understand and compare major modes in the

hydrographical and ecosystem components of the coupled marine system, namely for the mean winter (January-March) current field and net annual primary production, and to statistically compare these modes to potential driving environmental variables.

The method is comparable to the one used in Daewel et al., (2015), who gave the following brief introduction into the main elements of the analysis to clarify the terms used in the analysis. "The annual values of the spatially explicit variable field

form a NxM matrix χ (N: number of years; M: number of wet grid points). The empirical modes are given by the K eigenvectors of the covariance matrix with non-zero eigenvalues. Those modes are temporally constant and have the spatially variable pattern $p_k(m=1,...,M)$ where k=1,...,K. The time evolution $A_k(t=1,...,N)$ of each mode can then be obtained by projecting $p_k(m)$ onto the original data field χ such that $\chi(t,m) = \sum_{k=1}^{K} p_k(m)A_k(t)$. In the following we will refer to $A_k(t)$ as the principal components (PC) and to $p_k(m)$ as empirical orthogonal function (EOF). The percentage of the

variance of the field χ explained by mode k is determined by the respective eigenvalues and is referred to as the global explained variance $\eta_g(k)$.

Before using the method to analyse the spatiotemporal dynamics of the field, the data were demeaned (to account for the variability only) and normalized (to allow an analysis of the variability independent of its amplitude). The identified modes





are not necessarily equally significant in all grid points of the data field. Thus, the local explained variance $\eta_{local,k}(m)$ could provide additional information about the regional relevance of an EOF mode and the corresponding PC in percent:

$$\eta_{local}{}^{k}(m) = \left[ 1 - \frac{Var\left(\chi(m,t) - p^{k}(m)A^{k}(t)\right)}{Var(\chi(m,t))} \right] \cdot 100 \quad , \qquad (1)$$

where $Var(X) = \sum_{t=1}^{N}\left(\overline{X} - X(t)\right)^{2}$ denotes the variance of the field X(t)."

Note, that in our study the data were additionally low pass filtered using a 5-year running mean prior to applying the method. The principal modes of the EOF analysis are purely mathematical and not necessarily related to dynamical processes or physically interpretably. However, a good resolution of characteristic spatial and temporal scales improves the potential for several dynamically relevant modes (Schrum et al. 2006b).

Subsequently to the EOF analysis the major PCs were compared through correlation analysis to equally low pass filtered

time-series of environmental variables to identify potential environmental indicators and underlying processes. A Pearson correlation coefficient was estimated and tested against a t-distribution to obtain a measure for significance (Storch and Zwiers, 1999). A list of tested environmental variables is given in table 1. These variables were averaged in time (see table 1) and space (North Sea and Baltic Sea respectively) prior to analysis.

| Name | Explanation | Source |
|---|---|---|
| AMO | Atlantic Multidecadal Oscillation (index for North Atlantic Temperatures) | https://www.esrl.noaa.gov/psd/data/timeseries/AMO/ (Enfield et al., 2001) |
| WNAO | Winter North Atlantic Oscillation | https://www.esrl.noaa.gov/psd/gcos_wgsp/Timeseries/NAO/ (Hurrell, 1995) |
| Wind Speed | Average wind speed | NCEP/NCAR (Kalnay et al., 1996) |
| West-W | West-east wind component | NCEP/NCAR |
| East-W | East-west wind component | NCEP/NCAR |
| North-W | North-south wind component | NCEP/NCAR |
| South-W | South-north wind component | NCEP/NCAR |
| SWR | Short Wave Radiation | NCEP/NCAR |
| Airtemp | 2m air temperature | NCEP/NCAR |
| Precip | Precipitation | NCEP/NCAR |
| W-Winter | Average wind speed (Jan-Apr) | NCEP/NCAR |
| W-Summer | Average wind speed (May-Aug) | NCEP/NCAR |
| U-surf | Surface U- velocity component | ECOSMO |
| U_Winter | U- velocity component (Jan-Mar) | ECOSMO |
| V-surf | Surface V- velocity component | ECOSMO |
| V_Winter | V- velocity component (Jan-Mar) | ECOSMO |
| W-surf | Surface vertical velocity component | ECOSMO |
| W_Winter | Vertical velocity component (Jan-Mar) | ECOSMO |
| Current-speed | Average current speed | ECOSMO |
| SST | Seas surface temperature | ECOSMO |
| SSS | Sea surface salinity | ECOSMO |
| $NO_3$-surf | Surface $NO_3$ concentration | ECOSMO |
| $PO_4$-surf | Surface $PO_4$ concentration | ECOSMO |
| MLD | Average mixed layer depth | ECOSMO |
| MLD_May | Average mixed layer depth (May) | ECOSMO |

**Table 1. Variables used for correlation analysis with principle components of the net primary production EOF analysis (Fig. 7&8). Both atmospheric and oceanic variables were average over the respective sub-region (North Sea/Baltic Sea) for the analysis.**

### 2.3 Scenario simulations - Design

Three types of scenarios where designed to target the specific hypothesis deduced form the statistical analysis of model

results and previously published hypothesis on processes behind ecosystem changes in the North Sea and Baltic Sea (see



introduction). Here, we tested i) the impact of short wave radiation as a parameter determining the season length and intensity of the annual primary production, but also plays a role for changes in water temperature and mixed layer depth (MLD), ii) the impact of the wind forcing, which affects not only the general current field and nutrient supply from the open ocean to the North Sea, but also vertical mixing and upwelling, and hence mixing of nutrients to the euphotic layer, and iii)

the ecosystem response to changes in the river nutrient loads.

Instead of just increasing or decreasing the magnitude of the forcing parameters by a certain percentage, we aimed at resolving the impacts of the multi decadal variations for major shifts in the ecosystem dynamics. First analysis identified that the 60 years simulation period covered two different 30 year periods, for which productivity was significantly different (Daewel and Schrum, 2013). To identify the driving mechanisms for this change we divided the 61-year simulation period

into two climatic sub-periods (TP1: 1948-1976 & TP2: 1980-2008). Two climatic forcing variables were tested, SWR (sr) and wind stress (wi). For each of these two, scenario simulations were performed, for which all forcing variables but the target variable were kept unchanged. For the target variable, the forcing was repeatedly employed for both sub-periods (Fig. 3) such that in simulation 1 (sr1/wi1) the forcing from the TP1 was repeated in TP2 and in simulation 2 (sr2/wi2) the forcing from TP2 was also applied to TP1.

For the third set of scenarios we estimated average seasonal cycles for the river nutrient loads ($NO_3$,$PO_4$,$SiO$) in each of the 6 decades (Fig. 4) and performed a set of 6 simulations each forced by a different river load climatology. This enables us exploring the relevance of different persistent nutrient load situations and its relevance for abrupt changes in the system. The scenarios chosen include relatively high (80-90), intermediate (90-00) and low (00-08) nutrient loads, but also unusual N/P ratios in the forcing (70-80).

## 3 Results

### 3.1 Environmental indicators

To identify key long-term variations occurring in the North Sea and Baltic Sea system, we first investigated spatial averages of temperature, salinity and current speed for key regions. We focus here exemplary on the variations in the North Sea and present analysis in upper and lower water layer for the northern and southern North Sea respectively (Fig. 5&6). Our

analysis highlights several key characteristics related to long-term variations of hydrodynamics in the North Sea. Specifically, we find the following: An increase in temperature since beginning of the 90s was simulated for both northern and southern North Sea SST and bottom water temperature (Fig. 5). In the southern North Sea trends in surface and bottom layer are similar. However, this is not the case in the northern North Sea where temperature varies independently for surface and bottom waters. Substantial multi-year variations are superimposing the long-term trends in the North Sea temperature

and are evident in both surface and bottom layer. Additionally, surface water temperatures are also characterized by intra-annual variations. While, in the shallow southern North Sea the latter variations are also shown for the bottom layer, indicating a stronger coupling between surface and bottom in that region, the bottom layer of the deeper northern North Sea is largely uncoupled from these intra-annual variations. Also salinity patterns are dominated by long-term and decadal oscillations, whereof no long-term trend but rather multi-decadal variation is found in the northern North Sea. The southern

North Sea, in contrast, features an increasing trend in surface salinity, accompanied by a slightly weaker increase in bottom water salinity. Multi-year variations in salinity are comparable to those of temperature, but the strong intra-annual variability in surface temperature is not similarly evident for salinity and inter-annual and decadal to multi-decadal variability dominates. Current speed in the North Sea (Fig. 6) is dominated by a multi-decadal sinusoidal variation with low current speeds in the first 3 decades of the simulation period and higher current speed in the later 3 decades. A contrasting trend is

however found for the northern North Sea bottom layer showing a period of minimum current speed in the intermediate simulation period (1970-1990). Again here, a strong coupling between variability in surface and bottom layer is identified.





The potential of statistical analysis to provide more detailed information on long-term variations in North Sea and Baltic Sea currents is explored through EOF analysis of current vectors. In figure 7 we present the mean (averaged over the 61 year time period) surface current field in the North Sea and Baltic Sea, and the dominant mode from an EOF analysis over the anomalies to the mean current vector field for the winter season. The analysis indicates a substantial winter inflow anomaly

in the North Sea with current speeds from northwest to southeast during the last two decades. Contemporaneously the Baltic Sea was characterized by a substantial cross basin circulation anomaly from the Swedish towards the Polish coast that was likely related to a substantial ventilation of the Baltic Sea and nutrient transport from the lower layers to the euphotic zone as a consequence of enhanced coastal upwelling. This nutrient enhancement in the surface would foster the Baltic Sea primary production, a development that was indeed modelled (compare Fig. 10I and explanation below). Additionally we find

substantial decadal variability in the circulation. The first EOF thereby covers a significant part of the overall variability with more than 60% explained global variance. A similar analysis performed for current speed reveals that these anomalous current vectors are connected with a comparable pattern (though not identical) in increasing current speeds (Fig. 8c). The local explained variance of this analysis (Fig. 8b) shows that the pattern of increasing current speed (Fig. 8a) is particularly relevant in the central and north/north-western parts of the two main areas in the coupled North Sea and Baltic Sea system,

but does not explain variations of current speed in the southern and eastern coastal regions nor in the Bothnian Bay and Gulf of Finland areas.

### 3.2 Ecosystem variability

As highlighted above, changes in environmental variables are hypothesised to play a crucial role in explaining long-term changes in North Sea and Baltic Sea ecosystem dynamics. Here, we aim at identifying hydrodynamic and atmospheric

indicators, which could serve as a potential predictor for spatially resolved primary production changes. A number of indicators were tested, covering large-scale climate, regional atmospheric and regional hydrodynamic indicators. The predictive potential of these indicators was tested and comparatively assessed through correlations to the major principle components of primary production estimates (Fig. 9 & 10).

In the North Sea the first and second EOF explain the variability in the central North Sea and in the southern frontal areas

respectively (Fig. 9I&II), featuring substantially different temporal variability ($PC_1$ & $PC_2$). While in the central North Sea a major shift in primary production was simulated at around 1980 ($PC_1$), the production in the frontal regions passed through two major changes (around 1970, and around 1990) ($PC_2$). In general the signals ($PC_1$&$PC_2$) were overlaid by a quasi-decadal variability, which is comparable but not identical (partly caused by the statistical filtering procedures) to the variability estimated for the wind field.

The correlation analysis (Fig. 9III) reveals that the potential indicators for production are very different for the two patterns (relevant in the different sub regions). For the central North Sea, for which variability is mainly described by the first principal component ($PC_1$), changes in the **NAO**, changes in **wind speed,** specifically the western and southern wind component and, associated to it, in **current speed** show highest correlations to the major mode of variability in primary production, although several other variables are also significantly (at the 5% level) correlated to $PC_1$ (including SWR, winter

vertical velocity, surface salinity, $PO_4$ and $NO_3$). The production changes in the frontal area, in contrast, are significantly (at the 5% level) correlated only to 11 of the 25 considered environmental variables. Highest correlations can be found for the **AMO**, **air temperature, and precipitation** and, on the oceanic side, **SST** and the stratification index early in the season **MLD_May**. Despite the difference in regional and temporal variability, in both cases the most significant indicators are linked to processes driving the surface nutrient concentration, which is meaningful in a system where upper layer primary

production is limited by nutrient availability. Here, two regions are separated: i) The long-term variability in the seasonally



stratified central North Sea is mainly related to wind stress, which determines the nutrient inflow from the North Atlantic to the North Sea on the one hand but also impact vertical mixing and nutrient supply to the surface layer. ii) In the frontal areas off the Danish and English coast and at Dogger Bank the long-term changes in primary production are negatively correlated to the AMO, air temperature and precipitation, two parameters that impact the strength and timing of the seasonal

stratification. Here the effect is inversely proportional, the warmer the temperatures the stronger the stratification. Especially in regions with intermediate depths, a strong stratification and an early onset of the latter could substantially limit the nutrient supply to the euphotic zone.

In the Baltic Sea, almost 70 % of the overall simulated variability in primary production is described by the first EOF mode and PC (Fig. 10I). Here, we see a clear increase in primary production for the time period 1950-1987 and an abrupt increase

thereafter followed by an ever so slight decrease in primary production. The steep increase at the end of the 1980s has been shown to differentiate two statistically significant different periods (Daewel and Schrum, 2013) and clearly corresponds to the earlier described time for a regime shift in the Baltic Sea (Alheit et al., 2005). Daewel and Schrum (2013) showed that significant changes were evident for all three phytoplankton functional types, but that changes in cyanobacteria and flagellate production contributed mostly to the overall change. Hence, it is not surprising that surface $PO_4$ shows the highest

correlation (R=0.97) to the production change (Fig. 10III) and thus processes impacting the latter must play a significant role for primary production in the Baltic Sea. Nonetheless, in contrast to the North Sea, the correlation analysis for the Baltic Sea $PC_1$ did not indicate a dominant factor or process that could serve as an environmental indicator for production, since most of the considered parameters were found to significantly correlate to the main temporal changes in primary production (Fig. 10III). Additionally to the winter **NAO**, both **wind speed** and **SWR** are highly correlated to the major production pattern

($PC_1$). In contrast, the AMO was one of the few parameters with no significant correlation. The second EOF is less distinct, and explains only about 6% variability mostly in some coastal areas and in the Gulf of Bothnia (Fig. 10II). For the related $PC_2$ no clear relationships could be identified.

### 3.3 Causal Relationships

Since correlation analysis can identify statistical relations but not causality, we compiled subsequent scenario experiments

with the model to identify the role of variations in wind speed, SWR and river nutrient loads for production changes in the North Sea and Baltic Sea. Those parameters were chosen due to the high correlation we found between primary production and dynamic variables related to wind field changes (wind speed, wind components, current speed) and short wave radiation. The latter showed particular high correlation to Baltic Sea production variability. River loads were earlier hypothesized as one of the most relevant factors responsible for Baltic Sea system state changes from the late 1960s onwards (Thurow, 1997)

and for production changes in the southern North Sea (Clark and Frid, 2001). In figure 11 average low pass filtered time series for net primary production in the North Sea (southern North Sea and Northern North Sea) and Baltic Sea (central Baltic Sea and Gulf of Finland/Gulf of Riga) respectively are shown for the reference simulation and for the different scenario simulations. What becomes evident from this comparison is that the SWR forcing, although highly correlated to the Baltic Sea productivity and, besides nutrient availability, one of the main limiting factors for primary production, changes

surprisingly little of the low frequency variability in both North Sea and Baltic Sea productivity. Despite some small changes in short-term variability, especially in the southern North Sea, the multi-decadal variability and the major shifts remain unchanged in both areas. The wind forcing, on the contrary, can clearly be hold responsible for structuring the long-term variation. Most notably, our results indicate that the appearances of major shifts in the system (around 1980 in the North Sea and at the end of the 1980s in the Baltic Sea) are mainly caused by changes in the wind field, while the quasi-decadal

variations in the signal seems to remain largely unchanged. Note that we cannot exclude that the quasi-decadal variations in the newly compiled wind scenarios are coincidentally in phase with the variations in the reference forcing and hence, this



finding is no indication that the quasi-decadal variability is not attributed to wind field variations. However, in both areas the regime shifts in productivity are eroded or shifted in time when an alternative wind forcing is applied. This becomes most evident in the northern North Sea and in the central Baltic Sea, where the long-term production variability quite closely follows the variability in the wind field and sea surface current speed (compare also Fig. 5 & correlation analysis in Fig. 8&9), and the major shift e.g. in experiment wi1 is displaced to the end of the 1990s following the wind forcing dynamics from the TP1. Similar to the SWR experiments, a variation in the river nutrient loads does not change the long-term variability in ecosystem productivity substantially in neither the North Sea nor the Baltic Sea. However, it is shown that river loads clearly have an impact on the magnitude of the production in all areas, but especially in the Gulf of Finland and Gulf of Riga that both feature major river inflows. Clearly nutrient loads from the 1980s are highest resulting in higher system productivity. The comparison to the reference run shows that the river nutrient forcing does not cause major shifts in ecosystem productivity, but can clearly amplify changes in the system as seen in the two North Sea regions, where the production increase in the beginning of the 1980s is substantially enhanced by the high river nutrient loads in that decade. Interestingly, in the central Baltic Sea this effect is not similarly apparent. Here changes in nutrient loads aggregate and result rather in lower or higher production with the changes increasing slowly over time.

## 4 Discussion and Conclusion

We identified long-term multidecadal variations in temperature, salinity, currents and primary production in the North Sea and Baltic Sea from a coupled biological physical model simulation (Daewel and Schrum, 2013). While Daewel and Schrum (2013) already identified multi-decadal changes in simulated long-term dynamics of ecosystem productivity in the North Sea and Baltic Sea, the causes and underlying processes where only speculated on in their paper. One of the major advantages of coupled ecosystem models is the availability of all information relevant for the system dynamics including physics and forcing variables and so, underlying process interactions can be obtained via statistical analysis and scenario simulations.

As already shown by Janssen et al. (2001) the model is able to simulate long term dynamics in physical parameters. In this study we investigated exemplarily for the North Sea system average long-term changes in temperature, salinity and current speeds. Also here we find the long-term dynamics in temperature and salinity to cover average variability in observed temperature (Edwards et al., 2010) and salinity, by e.g. representing the "Great salinity anomaly" as observed between 1977-1981 in the North Sea (Danielssen et al., 1996). Besides temperature and salinity, current fields have been hypothesised to play a dominant role in ecosystem functioning. Here, average surface current fields for the northern and southern North Sea were identified to follow a similar long-term dynamics with a clear increase in current speed starting already in the beginning of the 1970s. This pattern is a result of the changing wind forcing above the North Sea as shown by Siegismund and Schrum (2001) who reported an intensification of west-south-westerly wind directions, an almost linear increase in wind speed and a more frequent appearance of "strong wind" events since the early 1970s. The same authors reported "an extension of winterly wind climate towards February and March during the last (analysed) decade (1988-1997), with pronounced preferences for west-southwesterly wind directions". A comparable mode of variability could be identified for the winter current vectors when analysed using EOF analysis. Here, both sub-regions (North Sea and Baltic Sea) have been analysed together, resulting in a mutual mode of variability that shows corresponding changes in winter current field anomalies after 1988 (compare Fig. 7&8). Mathis et al., (2015) published an EOF analysis for vertically aveeraged North Sea current velocities in winter (Dec/Jan/Feb.) simulated over the time period 1960-2000. Although the mean current field is not directly comparable to the surface currents analysed in this study, Mathis et al. (2015) concluded similarly on the impact of the westerly wind component for the inter-annual variability in the current field and circulation pattern. Also he found the changes in the circulation to be highly correlated to changes in the NAO. Their analysis showed that under stronger and more frequent westerly wind conditions the North Sea inflow through the Fair–Isle Passage was particularly enhanced




fostering a stronger southwards flow of Atlantic water masses along the British east coast. Under opposing weather conditions, the circulation in the central and southern North Sea weakens and the inflow through the Fair–Isle Passage follows the Dooley Current and, in that way, "effectively decoupling the water masses of the central and southern North Sea from the northern inflow" (Mathis et al., 2015). This process proofs especially relevant for the central North Sea, which is, in

contrast to well-mixed areas of the southern North Sea, neither strongly exposed to water inflowing from the English Channel nor to river runoffs, and can hence serve as an explanation for the provided correlation between the first mode in North Sea primary production variability and the NAO and wind field.

Applying EOF analysis to primary production allows identifying major modes of variability and their pattern together with a local indicator of explained variance. Here, the North Sea and Baltic Sea analysis lead to very different results. While in the

Baltic Sea we found one dominant mode that explains 67 % of the overall variability in primary production, the North Sea variability is spatially more diverse and we could identify at least two dominant modes of variability linked to specific spatial hydrodynamic features of the North Sea as described in Otto et al. (1990). Although, commenting on the occurrence and relevance of actual regime shifts in the North Sea and Baltic Sea is beyond the scope of our model, the estimated primary production analysis indicated indeed major "shifts" for the times when "regime shifts" have been identified in the

literature (e.g. Dippner et al., 2012; Weijerman et al., 2005), hence our findings can be considered relevant for explaining major indicators for RSs in the area. Clearly the results from our study indicate that analysing long-term variability of ecosystem dynamics for an average North Sea system is not sufficient. From the "regime shifts" detected in the North Sea, the change in 1978/1979 appears dominantly in the central North Sea (as indicated by the dominant mode of variability), while the second mode, relevant in the southern North Sea frontal areas, would at least show a stronger decrease in primary

production around 1990 where the second "regime shift" is presumed. While the second mode was correlated to air temperature and precipitation, environmental variables that affect the oceanic mixed layer depths, the first mode is clearly correlated to changes in the wind and current field and resembles the variability in average seas surface currents (compare figure 6 and explanation on North Sea circulation). As already described above, the main processes relevant for low frequency variations in primary production of the North Sea and Baltic Sea are specifically those impacting nutrient supply

in the euphotic zone. Although this is in line with what has been reported or the dynamics of the 78'/79' RS in Dippner et al. (2012), the variability for the central North Sea was, in contrast to their explanations, not correlated to the AMO nor to changes in the air temperature. Neither would our results support the hypothesis that changes in salinity (Lindeboom et al., 1995) nor changes in sunspot activity (results not shown) (Weijerman et al., 2005) caused changes in ecosystem dynamics. However, the identification of indicators for long-term variation assumes a priori that the indicator remains relevant for the

entire time period, while "regime shift" tailored studies usually do not consider the impact on the long-term dynamics and hence might come to different results.

The Baltic Seas primary production dynamics was almost in the entire basin linked to changes in the wind field. This was particularly evident from the performed scenario runs showing that, although nutrient loads would alter the magnitude of the primary production, the wind fields determine the timing and magnitude of long-term variations. In Daewel and Schrum

(2013) we already pointed out that the production variability is mainly seen in the flagellates and cyanobacteria bloom, while the here presented analysis indicate linkage to the winter current field (compare Fig. 7&8). In principle the underlying process can be explained by the 'cause-and-effect' chain proposed by Janssen et al. (2004) and the preconditioning of the deeper water column phosphate concentrations through eutrophication and anoxic conditions (Rodhe et al., 2006), which is additionally mediated by atmospheric conditions (Schinke and Matthäus, 1998). Such, our results would support the

hypothesis that long-term changes in primary production of the Baltic Sea are a consequence of eutrophication, even though the latter does not serve as an respective indicator for abrupt regime shifts. A similar argument has been formulated in the "regime shift" analyse by Österblom et al. (2007).





Here, we can conclude that changes in the wind speed and/or changes in the east-west component of the wind field, can serve as an indicator or maybe even as a predictor for changes in primary production in both targeted areas. Even in the southern North Sea the changes in wind fields explain more of the long-term production changes than variations in the nutrient forcing, which would, at least partly, contradict conclusions from Clark and Frid (2001) on the southern North Sea

phytoplankton dynamics.

However, it need to be pointed out that this analysis is performed to identify indicators for low frequency variability, correlations are substantially weaker on un-filtered time series. Moreover, climatic conditions might change and the relevance of specific processes for inter-annual changes in production can alter due to changes in environmental and climate conditions. An example from our model study are variations in North Sea nutrient loads, which caused an amplification of

the wind induced variations in the 1980s in the northern North Sea as well as alterations of the primary production variability in the southern North Sea after 1990 when nutrient loads were substantially reduced. Other possible examples are changes in stratification and, at least in the Baltic Sea, sea ice retreat that could cause variations on primary production and become more relevant under future climate, in which case air temperature or short wave radiation could become a more significant indicator than wind speed.

**Acknowledgements**

This work is a contribution to the FP7-SeasERA SEAMAN Collaborative Project financed by the Norwegian Research Council (NRC-227779/E40).

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





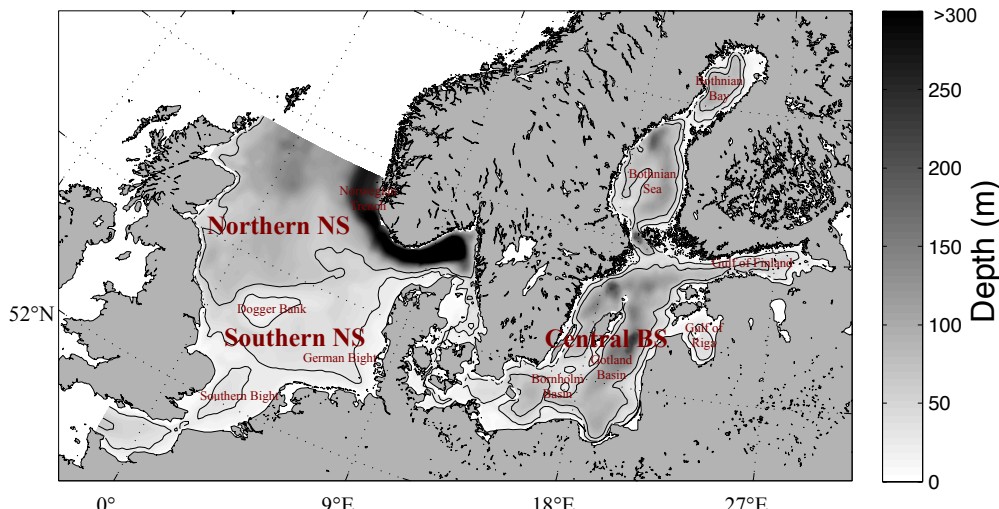

Figure 1: Model area and bathymetry. Black lines indicate the 30 m and 60 (the 60 m depth line separates northern and southern North Sea; Central BS includes all areas east of 14°E excluding the gulf regions) m depth lines respectively.



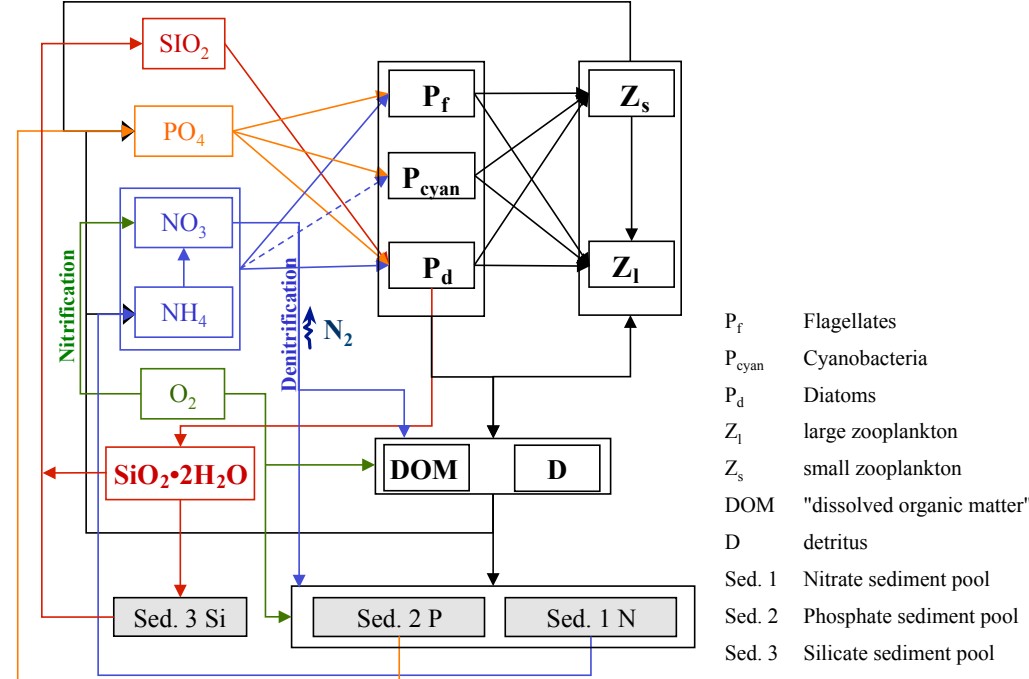

**Figure 2: Schematic diagram of biochemical interactions in ECOSMO II (Daewel and Schrum, 2013).**





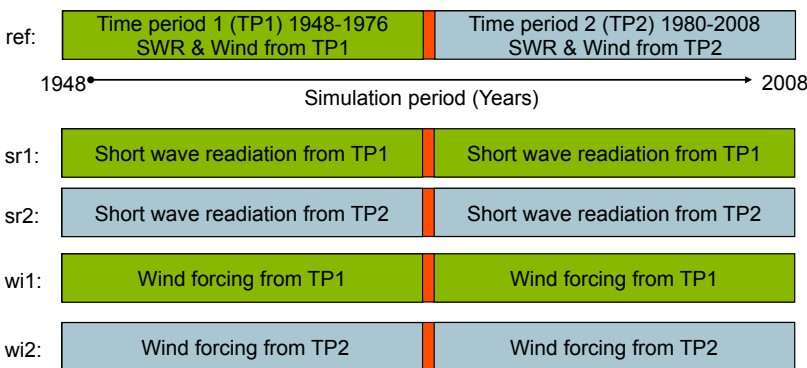

**Figure 3: Schematic diagram for the scenario simulation setup. The setup is valid for the short wave radiation experiments (sr1/sr2) and for the wind experiments (wi1/wi2), ref denotes the reference simulation from Daewel and Schrum (2013).**



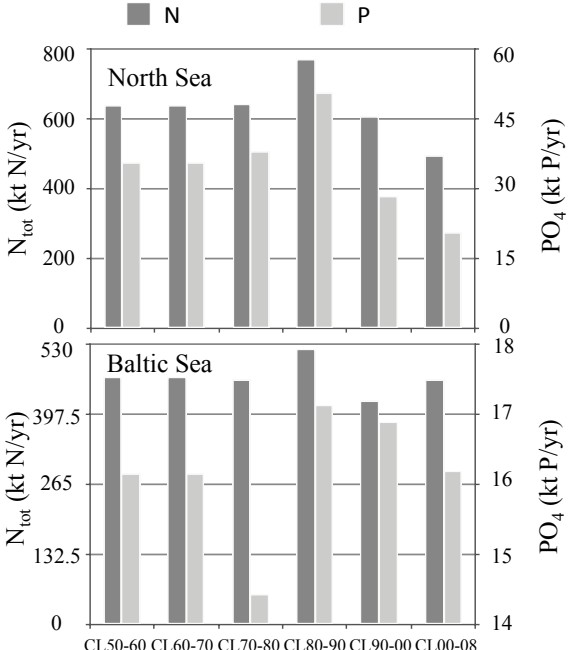

**Figure 4: Decadal mean annual nutrient loads $N_{tot}$ ($NO_3 + NH_4$) and $PO_4$ averaged for each of the 6 simulation decades for use in the scenario simulations. Note: SiO has also been modified but is not shown here.**





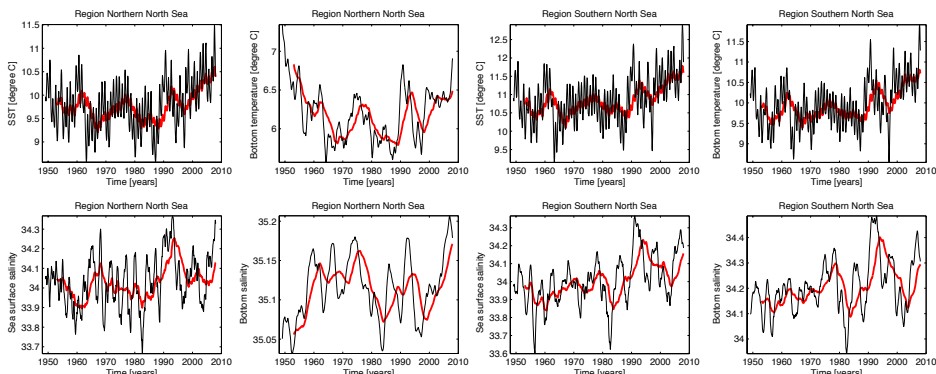

**Figure 5: Northern North Sea (left two columns) and Southern North Sea (right two columns) temperature (upper) and salinity (lower) in surface (left) and bottom layer (right). Displayed are monthly data as 13pt. moving average (black) and 61pt moving average (red).**





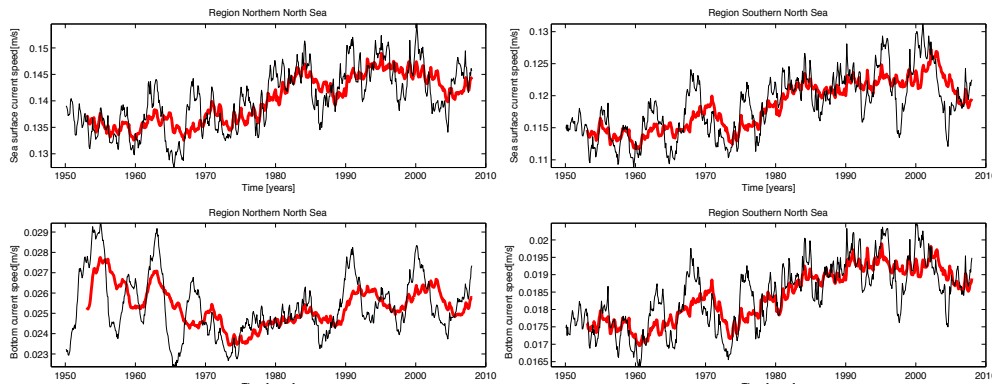

**Figure 6: Northern (left) and Southern (right) North Sea surface (upper) and bottom current speed (lower). Displayed are monthly data as 25pt. moving average (black) and 61pt moving average (red).**





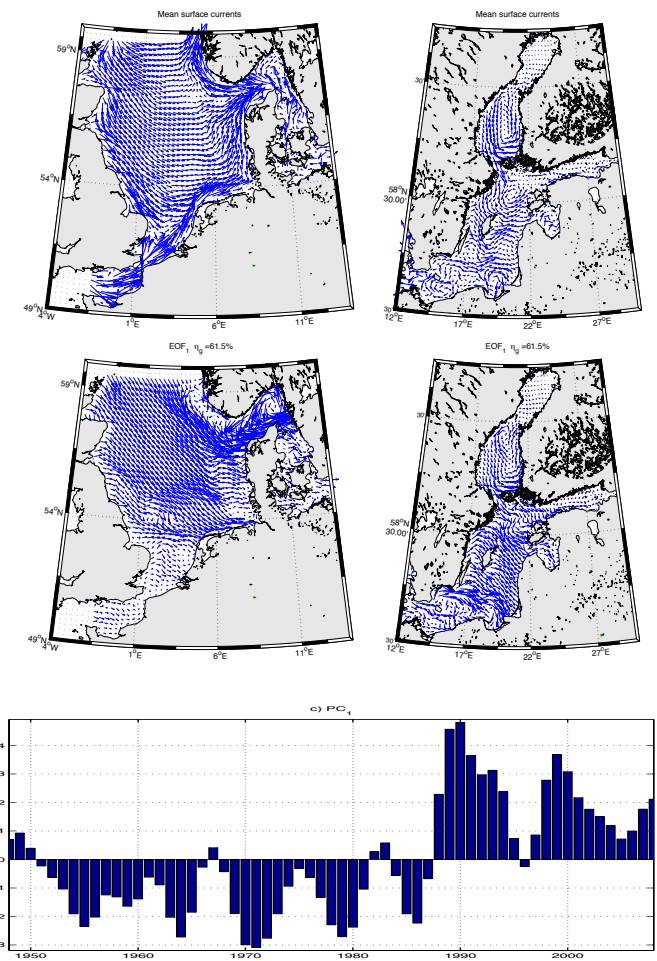

**Figure 7: Mean surface current vectors in North Sea (upper left) and Baltic Sea (upper right), EOF analysis of the anomalies in current vectors for the winter period Jan-March: current pattern for the first EOF (middle) and first principle component (lower).**



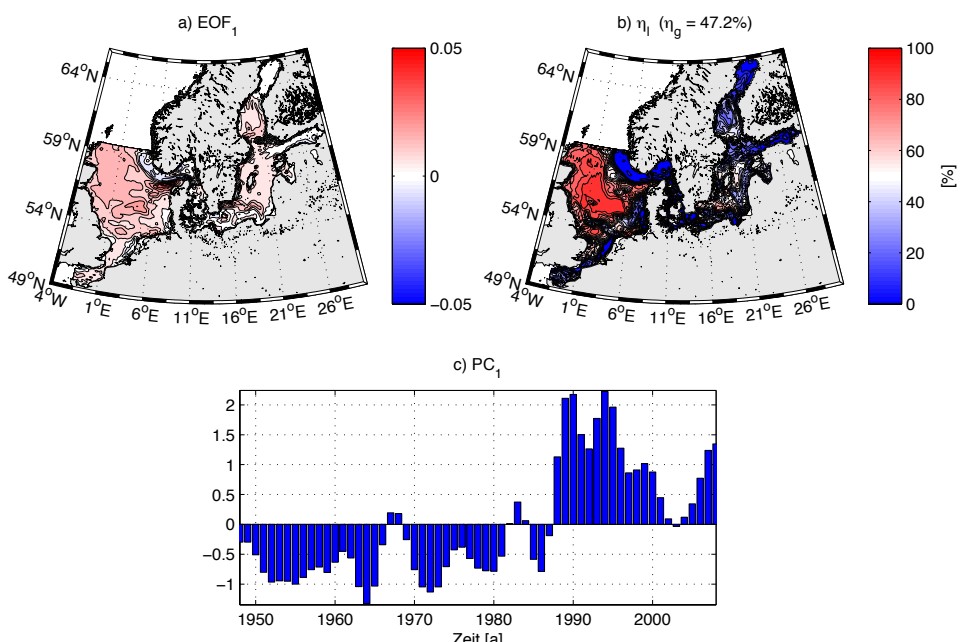

**Figure 8:** EOF analysis of the anomalies in current speed for the winter period Jan-March, current speed pattern
for the first EOF (upper left), local explained variance (right) and first principle component (lower).

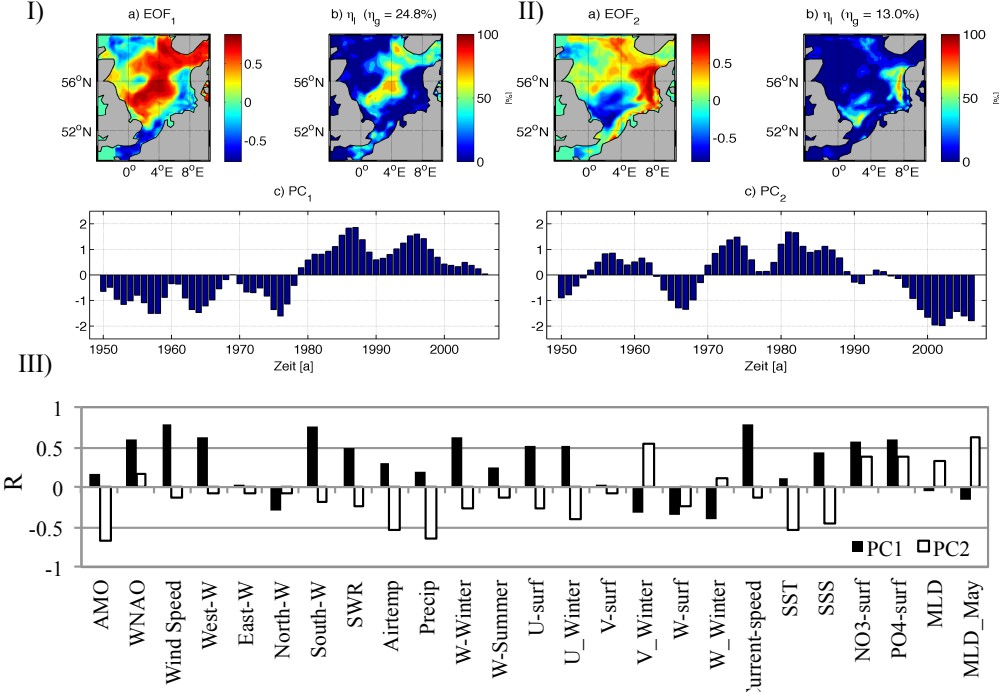

Figure 9: I&II) a) First and second empirical orthogonal function for annual mean primary production in the North Sea (1948-2008); b) local explained variance for the pattern for the corresponding EOF; c) principle component (time variation) of the corresponding EOF. III) absolute values of the correlation coefficient between the principle components (PC1 & PC2) and an environmental variable stated on the x-axis.





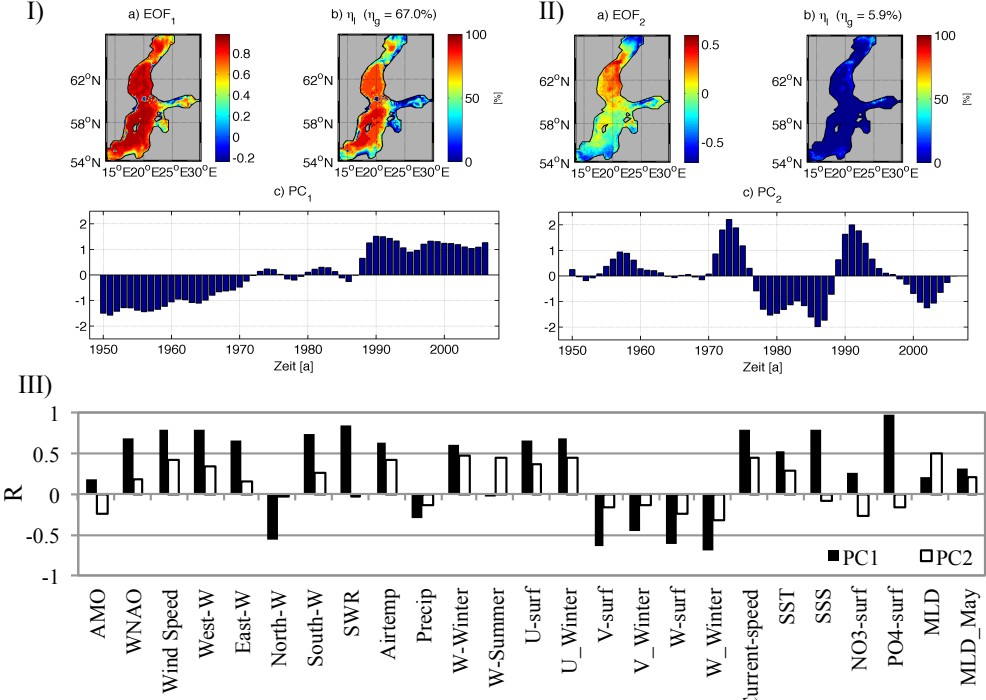

**Figure 10: I&II) a) First and second empirical orthogonal function for annual mean primary production in the Baltic Sea (1948-2008); b) local explained variance for the pattern for the corresponding EOF; c) principle component (time variation) of the corresponding EOF. III) absolute values of the correlation coefficient between the principle components (PC1 & PC2) and a environmental variable stated on the x-axis.**



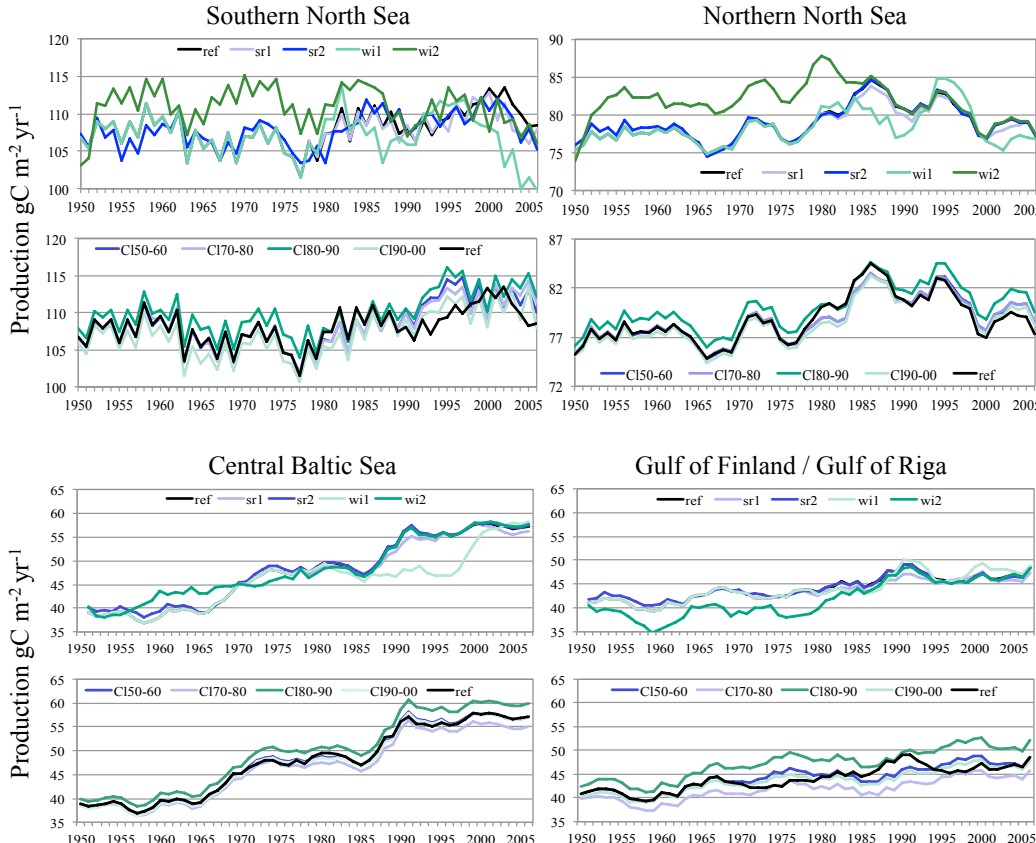

**Figure 11: Estimated net primary production for the reference run (ref) and the scenario simulations concerning short wave radiation (sr1/sr2) and wind (wi1/wi2) (upper pannels) and river nutrient nutrient load (Cl) (lower panels) for two subregions in the North Sea (southern & northern North Sea) and two subregions in the Baltic Sea (central Baltic Sea & Gulf of Finland / Gulf of Riga)**