# Peer review of "Low frequency variability in North Sea and Baltic Sea identified through simulations with the 3-d coupled physical-biogeochemical model ECOSMO"

_Earth System Dynamics, 2017_

## Referee Comment (RC1) · Anonymous Referee #1 · 8 May 2017

General Remarks:

The paper presents results from a 61-year long hindcast simulation using a coupled hydrodynamic-ecosystem model applied to the North Sea and Baltic Sea. By means of this model system, the long-term multi-decadal variability of relevant physical and biological parameters is investigated. In addition, specific sensitivity tests have been performed in order to determine the role of single forcing functions on the detected variability.

The overall impression is that the paper is very carefully written in a clear and concise

way. In particular, the sensitivity tests are nicely chosen to illustrate the impact of different forcing functions on the low frequency variability of physical and ecosystem related parameters. Only for some of the results, the descriptions should be expanded a little to make it easier for the reader to follow. More details and some other minor comments are given below. In summary, I can recommend the manuscript for publication after a minor revision.

Detailed Comments:

Page 2, line 22: Please be consistent throughout the text: "long term" or "long-term".

Page 2, lines 26-27: What is meant by "causal relations to inter-annual variations"?

Page 3, line 21: Please add "spatially and temporally explicit character . . ."

Page 3, line 21: It is misleading to say that the explicit character of model data make it difficult to find major variability modes. Compared to observational data, only such an explicit character makes it possible to perform a thorough analysis.

Page 3, line 29: Quotation marks at the end of the block are missing.

Page 4, lines 7-8: I do not see a general link between spatial and temporal resolution and better chances that EOF modes are related to "real" physical modes. According to my understanding the potential that specific physical processes can be represented by orthogonally arranged EOFs, is not necessarily connected to the resolution. Also, in the cited reference Schrum et al. (2006b), no information could be found, which supports this statement. The reference seems to be misplaced here. Therefore the authors should reconsider this sentence.

Page 5, line 8: Why "60-years" simulation period? Everywhere else, 61 years are stated.

Page 6, line 2: Scale and units are missing for current vectors in Fig. 7.

Page 6, line 11: In principle, Fig. 7 already shows the current speed in form of the

vector length. Hence, the sentence should be reformulated.

Page 6, lines 15-16: From Fig. 8a, it is not clear, whether fluctuations could not be explained or whether they are not present at all.

Page 7, lines 2-7: It would be much easier to follow this paragraph, if the specific EOFs, which are referred to and discussed in the text, are mentioned.

Pages 7 and 8: Section 3.3: This section should be extended a little in order to make the results more clear. In particular, when mentioning the different scenarios, it would be helpful if the idea behind the specific scenarios is briefly repeated in a half-sentence.

Page 8, line 39: Also THEY (Mathis et al.) found out . . ..

Figs. 8 and 9: The figures showing the horizontal EOF distribution in I) and II) are too small to identify all the important features.

–––––––––––––––––––––––––––

---

## Referee Comment (RC2) · Anonymous Referee #2 · 9 May 2017

This manuscript is about a statistical evaluation of a numerical simulation of the coupled system North Sea and Baltic Sea. The model used is a 3D ocean model coupled with a marine biogeochemical model. The model simulation spans the period 1948-2008. EOF technique has been used as the statistical method. Main aim of the study was to identify environmental drivers causing low frequency variability and to explore possible cause-effect relationships.

The experimental setup is adequate for the research questions. The manuscript is well written and concise, and conclusions are drawn in a comprehensive manner. I would

recommend a publication after a "minor revision".

I suggest a careful proof-reading of the text to eliminate inconsistencies, e.g. the use of inter-annual vs. interannual vs. intra-annual. Page 3, line 21: I cannot agree with this statement or something is missing.

Some figures need corrections: Fig. 4 axis label: I suggest to use "CL50-59 CL60-69 ..." Figs. 4,8,9,.. axis labels: Use time instead of Zeit.

---

## Author Comment (AC2) · 9 Jun 2017

We would like to thank the reviewer for taking the time and effort to review our work. The comments will be considered in an updated version of the manuscript. Specifically, we will carefully check the text for inconsistencies and revise the figures according to the suggestions.

Answer to the comment concerning Page 3, line 21: The reviewer is of course right, this is poorly phrased. We will change the text to:" The advantage of model-derived data

is their spatially explicit characteristics, which allows resolving the variability on various time and spatial scales. To identify major modes of variability we apply a widely used method in climate and ocean science, the empirical orthogonal function analysis, a statistical method . . . .

---

## Author Response (AR1)

Answer to reviewer 1

We would like to thank the reviewer for the effort spending on our manuscript and for the useful comments that will be taken into account in a revised version of the manuscript. When preparing a new version of the manuscript we will consider all comments suggested by the reviewer and specifically will revise the manuscript for consistency in the formulations and comprehensibility. Also we will revise the figures as suggested.
Our answers to the more extensive comments can be found below:

Page 2, lines 26-27: What is meant by "causal relations to inter-annual variations"?
**Answer:** Better is:„... to understand the causal relationship between indicators and the low frequency variability of simulated primary production."

Page 3, line 21: It is misleading to say that the explicit character of model data make it difficult to find major variability modes. Compared to observational data, only such an explicit character makes it possible to perform a thorough analysis.
**Answer:** The reviewer is of course right, this is poorly phrased. We will change the text to:" The advantage of model-derived data is their spatially explicit characteristics, which allows resolving the variability on various time and spatial scales. To identify major modes of variability we apply a widely used method in climate and ocean science, the empirical orthogonal function analysis, a statistical method ... .

Page 4, lines 7-8: I do not see a general link between spatial and temporal resolution and better chances that EOF modes are related to "real" physical modes. According to my understanding the potential that specific physical processes can be represented by orthogonally arranged EOFs, is not necessarily connected to the resolution. Also, in the cited reference Schrum et al. (2006b), no information could be found, which supports this statement. The reference seems to be misplaced here. Therefore the authors should reconsider this sentence.
**Answer**: In Schrum et al. (2006b) can be found: "It should be noted that identified modes of variability are by no means automatically related to dynamic processes as the statistical decomposition can fail in providing pattern which are dynamically relevant. However, in the case of a proper adaptation of regional and temporal windows with respect to the variability of the parameter investigated, the problem can be reduced and the separation into statistical modes helps to understand the relation between spatial and temporal variability and gives some hints to the dynamic processes responsible." We agree that the use of the word resolution might be misleading and suggest changing the sentence to: "However, the use of a proper regional and temporal window encompassing the potential scales of variability of the targeted parameter improves the potential for several dynamically relevant modes (Schrum et al. 2006b).

Page 6, line 2: Scale and units are missing for current vectors in Fig. 7.
**Answer:** Scale and units were added.

Page 6, line 11: In principle, Fig. 7 already shows the current speed in form of the vector length. Hence, the sentence should be reformulated.

**Answer:** Although the analysis in Fig. 7 describes the change in current speed through the length of the current vector, it only describes the change for this specific part ($EOF_1$) of the circulation. The second analysis Fig. 8, in contrast, describes the variability in the scalar current speed and hence gives additional information. We suggest changing the text to: "An additional EOF analysis performed for the scalar current speed further highlights the fact that this strong increase in strength of the northwest current component is connected to a general increase in current speed (Fig. 8c)."

Page 6, lines 15-16: From Fig. 8a, it is not clear, whether fluctuations could not be explained or whether they are not present at all

**Answer:** The local explained variance shown in figure 8b shows clearly that the first EOF of the analysis is not relevant in these areas. Meaning that this specific pattern of temporal variation, namely the increase in current speed, is not present there. There are of course fluctuations in those areas, but they are different from those explained by the first dominant mode of variability. We suggest adapting the text a bit to make this clearer: "The local explained variance of the first EOF mode (Fig. 8b) shows that this dominant mode of variability (Fig. 8a) is highly relevant in the central and north/north-western parts of the two main areas in the coupled North Sea and Baltic Sea system. However, it does not explain variability in the southern and eastern coastal regions nor in the Bothnian Bay and Gulf of Finland, indicating that the current speed variability in these areas differ substantially from the dominant pattern."

Page 7, lines 2-7: It would be much easier to follow this paragraph, if the specific EOFs, which are referred to and discussed in the text, are mentioned
**Answer:** Yes, right. We will consider this in a new version of the manuscript.

Pages 7 and 8: Section 3.3: This section should be extended a little in order to make the results more clear. In particular, when mentioning the different scenarios, it would be helpful if the idea behind the specific scenarios is briefly repeated in a half-sentence.
Answer: We suggest inserting another sentence to the text: "Since correlation analysis can identify statistical relations but not causality, we compiled subsequent scenario experiments with the model to identify the role of variations in wind speed, SWR and river nutrient loads for production changes in the North Sea and Baltic Sea. Those parameters were chosen due to the high correlation we found between primary production and dynamic variables related to wind field changes (wind speed, wind components, current speed) and short wave radiation. The latter showed particular high correlation to Baltic Sea production variability. River loads were earlier hypothesized as one of the most relevant factors responsible for Baltic Sea system state changes from the late 1960s onwards (Thurow, 1997) and for production changes in the southern North Sea (Clark and Frid, 2001). **To emphasize the changes in variability rather than magnitude, the temporal variability of the single forcing**

**parameters where modified as described in section 2.3 (see also figure 3).**
…"

Figs. 8 and 9: The figures showing the horizontal EOF distribution in I) and II) are too small to identify all the important features.
**Answer:** The figures were restructured a bit to allow larger sub-figures. The content of the figures remains unchanged.

Answer to reviewer #2

We would like to thank the reviewer for taking the time and effort to review our work. The comments will be considered in an updated version of the manuscript. Specifically, we will carefully check the text for inconsistencies and correct the figures.
**Answer:** The inconsistencies in the text as well as incorrect figure captions were corrected in the new version of the manuscript.

Page 3, line 21: I cannot agree with this statement or something is missing.
**Answer:** The reviewer is of course right, this is poorly phrased. We will change the text to:" 
[revised manuscript text omitted]